Subject Areas:
bioengineering/green chemistry/biotechnology

Keywords:
biocatalysis, L-amino acid oxidases, α-keto acid, amino acids, biotransformation

Author for correspondence:
Ziduo Liu
e-mail: ziduoliu_hzauedu@sina.cn

This article has been edited by the Royal Society of Chemistry, including the commissioning, peer review process and editorial aspects up to the point of acceptance.

# Efficient production of α-keto acids by immobilized E. coli-pETduet-1-PmiLAAO in a jacketed packed-bed reactor

Licheng Wu[1], Xiaolei Guo[2], Gaobing Wu[3], Pengfu Liu[4] and Ziduo Liu[3]

[1]Department of R&D of zhejiang zhengshuo Biological Co., Ltd, Huzhou 313000, Zhejiang, People's Republic of China
[2]College of life science, Fujian normal university, Fuzhou, Fujian 350000, People's Republic of China
[3]College of Life Science and Technology, State Key Laboratory of Agricultural Microbiology, Huazhong Agricultural University, Wuhan 430070, People's Republic of China
[4]Zhejiang University of Technology, Hangzhou 310014, Zhejiang, People's Republic of China

ZL, 0000-0001-6410-0420

α-keto acids are compounds of primary interest for the fine chemical, pharmaceutical and agrochemical sectors. L-amino acid oxidases as an efficient tool are used for α-keto acids preparation in this study. Firstly, an L-amino acid oxidase (PmiLAAO) from Proteus mirabilis was discovered by data mining. Secondly, by gene expression vector screening, pETDuet-1-PmiLAAO activity improved by 130%, as compared to the pET20b-PmiLAAO. PmiLAAO production was increased to 9.8 U ml$^{-1}$ by optimized expression condition (OD$_{600}$ = 0.65, 0.45 mmol l$^{-1}$ IPTG, 20 h of induction). Furthermore, The PmiLAAO was stabile in the pH range of 4.0–9.0 and in the temperature range of 10–40°C; the optimal pH and temperature of recombinant PmiLAAO were 6.5 and 37°C, respectively. Afterwards, in order to simplify product separation process, E. coli-pETduet-1-PmiLAAO was immobilized in Ca-alginate beads. Continuous production of 2-oxo-3-phenylpropanoic acid was conducted in a packed-bed reactor via immobilized E. coli-pETduet-1-PmiLAAO. Significantly, 29.66 g l$^{-1}$ 2-oxo-3-phenylpropanoic acid with a substrate conversion rate of 99.5% was achieved by correspondingly increasing the residence time (25 h). This method holds the potential to be used for efficiently producing pure α-keto acids.

# 1. Introduction

The α-keto acid is a compound containing both a carboxyl group and a ketone group in a molecule involved in the TCA cycle (tricarboxylic acid cycle, also well known as citric acid cycle), decomposition of amino acids and significant in biological systems [1]. α-keto acids and their derivatives (α-hydroxy acids) as a building block have been applied to chemical synthesis, dietary supplements, chemical industries and pharmaceutical (such as ibuprofen and naproxen, (S)-camptothecin, (S)-oxybutynin) [2]. For instance, D-phenylglycine was used for the preparation of HCV NS5A inhibitor [3] and β-lactam antibiotics (ampicillin and cephalexin) with a consumption of several thousand tons per year [4]. Especially, 4-fluoro-D-phenylglycine and 4-chloro-D-phenylglycine have been used for h5-HT1D receptor agonist [5] and macrocyclic hedgehog pathway inhibitor preparation [6], respectively. However, α-keto acids as a kind of unstable compound are difficult to synthesize, and are easily decarboxylated, decarbonylated and oxidized. Three different approaches have been developed for α-keto acids preparation: (i) chemical synthesis: complex raw materials, expensive catalysts and complicated processes; (ii) microbial fermentation: low yield, purification process is complex and costly and (iii) biotransformation. Biotransformation with high performance has been regarded as the core of green chemistry.

Biotransformation as a well-suited method has been applied to prepare α-keto acids and chiral rare amino acids, in which enzymes as 'green' biocatalysts have been used to produce fine chemicals and pharmaceuticals [7,8]. Amino acid oxidase (EC 1.4.3.2), also called amino acid deaminase (AAD), including L-amino acid oxidase (L-AAO) and D-amino acid oxidase (D-AAO), is a type of flavoenzyme with a broad spectrum of oxidase activity [9]. α-amino groups of amino acids can be removed by AAD in the presence of oxygen and produce α-keto acids and ammonia [10]. AAO can be produced by a variety of microorganisms, including *Trichoderma viride*, *Pseudomonas*, *Proteus mirabilis*, *Penicillium*, *Aspergillus niger* and *Aerobacter aerogenes* [11–13]. In past years, the D-amino acid oxidase (D-AAO) has been applied to prepare L-amino acids from racemic mixtures [14]. In this study, we are trying to obtain an efficient biocatalyst for α-keto acid production. To obtain high-yield, high-purity α-keto acids to meet the needs of industrial production, an amino acid oxidase from *Proteus mirabilis*, was selected by data mining and screening. Moreover, the amino acid oxidase genes have been cloned and the production of the LAAO was optimized by cultivation conditions. In addition, the production of α-keto acid has been enhanced by immobilization of *E. coli*-pETduet-1-*Pmi*LAAO in Ca-alginate beads and reaction in a packed bed reactor (scheme 1). This method holds the potential to be used for efficient production of pure α-keto acids.

# 2. Material and methods

## 2.1. Construction of recombinant strain

The corresponding LAAO genes were synthesized by the Sangon (Shanghai, China). The LAAO genes were then amplified by polymerase chain reaction (PCR) techniques and the used primers were shown as follow: *Pmi*LAAO-F (5′-ATGAACATTTCAAGGAGAAAGCTAC-3′) and *Pmi*LAAO-R (5′-ACTTC TTAAAACGATCCAAACT-3′); *Ro*LAAO-F (5′-ATGGCATTCACACGTAGATC TTTCA-3′) and *Ro*LAAO-R (5′-TCAGGCTTCCTGGGCCACG-3′) [5]; *Dr*LAAO-F (5′-AGTCTTCAAGCCAATAAG-3′) and *Dr*LAAO-R (5′-GGGAC TATAGCTCCTAGAAT-3′) [6]. Then the amplified genes were ligated into plasmid pACYCduet-1, pETDuet-1, PRSFduet-1, PCDFduet-1 and pET20b (Novagen, Germany), respectively, then verified by plasmid PCR and DNA sequencing. The recombinant plasmids were transformed into *E. coli* BL21 (DE3) for expression. The positive clones were cultivated in LB medium containing ampicillin (100 μg ml$^{-1}$) at 37°C, 200 r.p.m. Subsequently, recombinant *E. coli* BL21 (DE3) induced by supplementing of 0.45 mM IPTG for further 20 h at 16°C while the $OD_{600nm}$ was 0.6 approximately.

## 2.2. Purification of *Pmi*LAAO

After fermentation, the cells were collected by centrifugation at 6000 r.p.m. for 20 min. The cells were washed twice with 0.2 M phosphate buffer (PBS, pH 7.2) and suspended in 0.2 M PBS (pH 7.2). The cells were homogenized by ultrasonic cell disruptor (power 300 W, ultrasound 4 s, pause 8 s, total 30 min) under ice-cooling. The supernatant was obtained by centrifugation at 8000 r.p.m. for 10 min.

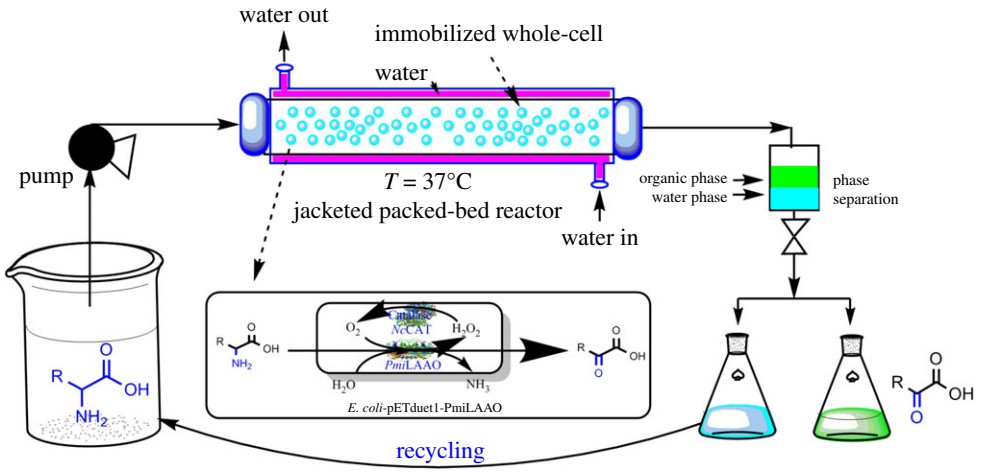

**Scheme 1.** Continuous-flow biosynthesis of α-keto acids from L-amino acid by immobilized whole-cell of *E. coli*-pETduet-1-*Pmi*LAAO. The pH of the aqueous phase containing unreacted amino acids (substrate) had been adjusted to 6.5 before recycling.

In order to remove those proteins with a molecular mass below 30 kDa, the supernatant was filtered by ultrafiltration with an Amicon Ultra-15 30 K device (Millipore, USA). The Ni(II)-NTA agarose matrix (QUIAGEN), a metal chelation chromatography, was used for further purification of the concentrate containing recombinant *Pmi*LAAO.

## 2.3. Measurement of the amino acid oxidase activity

The AAO activity was determined at pH 7.5 and 30°C with L-Phe as substrate according to the method of Sacchi *et al.* [15]. The initial production rate of hydrogen peroxide ($H_2O_2$) with a coupled peroxidase assay was measured. Briefly, the crude extracts of the L-AAO 100 µl were added with 100 µl testing solution (1 mmol l$^{-1}$ *o*-dianisidine (*o*-DNS), 20 mmol l$^{-1}$ L-Phe and 20 U ml$^{-1}$ horseradish peroxidase (HRP), in 0.2 mol l$^{-1}$ dipotassium pyrophosphate buffer, pH 7.5) cultivated 10 min at 30°C. The time course of the absorbance change at 440 nm was recorded by using UV/Vis spectrophotometer. The amount of enzyme required to consume 1 µmol of L-Phe per minute under the described conditions has been defined as one unit of LAAO activity.

## 2.4. The *Pmi*LAAO characterization

The optimum pH for *Pmi*LAAO was determined in the range of 4–11. pH-stability was investigated at different pH under 30°C. The optimum temperature of *Pmi*LAAO was determined in the range of 10–70°C. Thermo-stability was investigated at different temperatures in the 0.2 M phosphate buffer (PBS, pH 6.5). The residual activity of the samples was measured as described above.

## 2.5. Immobilization of *E. coli-Pmi*LAAO

*E. coli-Pmi*LAAO entrapment in calcium alginate beads. Sodium alginate aqueous solution (2.5%) was mixed with *E. coli-Pmi*LAAO whole-cell suspension (30 mg cells ml$^{-1}$) with a ratio of 1 : 1 (v/v). After that, in order to prepare Ca-alginate beads, the mixture was dropped into 0.3 mol l$^{-1}$ CaCl$_2$ solution using a syringe. Preparation of cell-carrageenan beads and gelatin beads were completed according to the Ca-alginate beads.

Cell cross-linking with glutaraldehyde was conducted as follows: *E. coli-Pmi*LAAO whole-cell was suspended in 0.1 M PBS buffer (pH = 7.0) at a concentration of 25 g l$^{-1}$. Subsequently, 2.5% (v/v) glutaraldehyde was mixed with *E. coli-Pmi*LAAO whole-cell suspension. To obtain a cross-linked preparation, the mixtures were incubated at room temperature (25°C) for 60 min. The cell cross-linked *E. coli* was obtained and the supernatant containing excess glutaraldehyde was removed by centrifuging for 10 min. Moreover, preparation of polyacrylamide gel beads was completed according to the protocol by Skryabin & Koshcheenko [16].

**Table 1.** Biocatalysts screening.

| no. | amino acid oxidases (LAAO)[b] | microorganism | Km (mM)[a] |
| --- | --- | --- | --- |
| 1 | *Ro*LAAO | *Rhodococcus opacus* | 0.022 |
| 2 | *Nc*LAAO | *Neurospora crassa* | 0.16 |
| 3 | *Pre*LAAO | *Providencia rettgeri* | 3.1 |
| 4 | *Dr*LAAO | *Daboia russellii* | 0.0665 |
| 5 | *Th*LAAO | *Trichoderma harzianum* | 11.73 |
| 6 | *No*LAAO | *Naja oxiana* | 0.051 |
| 7 | *Cr*LAAO | *Calloselasma rhodostoma* | 0.05 |
| 8 | *Cad*LAAO | *Crotalus adamanteus* | 0.03782 |
| 9 | *Cat*LAAO | *Crotalus atrox* | 0.036 |
| 10 | *Mg*LAAO | *Meleagris gallopavo* | 3.50 |
| 11 | *Hs*LAAO | *Hebeloma cylindrosporum* | 2.20 |
| 12 | *Pmi*LAAO | *Proteus mirabilis* | 22 |

[a]Data were collected from Enzyme Database-BRENDA (https://www.brenda-enzymes.org/).
[b]Data were obtained by pBLAST (https://www.ncbi.nlm.nih.gov/gene).

## 2.6. Preparation of α-keto acids from L-amino acid by whole-cell biocatalyst

The α-keto acids preparation reaction was shown as follows: $1 \, g \, l^{-1}$ L-phenylalanine or its derivative, whole-cell biocatalyst $2 \, g \, l^{-1}$ (wet cell weight, w/v), $0.2 \, mol \, l^{-1}$ phosphate buffer (PBS, pH 6.5) at 37°C, 200 r.p.m. The reactions were stopped by centrifugation at 8000 r.p.m. for 15 min. Then, the supernatant was withdrawn and the α-keto acids content was measured by high-performance liquid chromatography (HPLC) [4]; briefly, HPLC equipped with an AminexHPX-87H column at 35°C with the injection volume of 10 µl. The mobile phase was $5 \, mmol \, l^{-1}$ $H_2SO_4$, and the flow rate was $0.6 \, ml \, min^{-1}$.

# 3. Results and discussion

## 3.1. Discovery of L-amino acid oxidase

In order to construct a library of L-amino acid oxidases (LAAO) for α-keto acid production, the amino acid sequence of *Pmi*LAAO, *No*LAAO and *Mg*LAAO as a query were used for a pBLAST search. Twelve proteins possibly having L-amino acid oxidase activity from various microorganisms were selected by data mining. In addition, the Km value of the amino acid oxidases towards L-phenylalanine has been investigated (table 1). However, many of LAAO gene sequences with lower Km value (including *Nc*LAAO, *Pre*LAAO, *Th*LAAO, *Hs*LAAO and *Cat*LAAO) failed to be accessed from the GenBank database (https://www.ncbi.nlm.nih.gov/genbank/). *No*LAAO, *Cr*LAAO, *Cad*LAAO, *Mg*LAAO, *Pmi*LAAO, *Ro*LAAO and *Dr*LAAO with available sequence data were selected as the candidates. Afterwards, pET20b-*No*LAAO, pET20b-*Cr*LAAO, pET20b-*Cad*LAAO, pET20b-*Mg*LAAO, pET20b-*Pmi*LAAO, pET20b-*Dr*LAAO and pET20b-*Ro*LAAO were constructed and transformed into the expression host *E. coli* BL21 (DE3), subsequently. Moreover, the amino acid oxidase activity of the recombinant strains has been tested. Unfortunately, the *No*LAAO, *Cr*LAAO, *Cad*LAAO, *Mg*LAAO and *Dr*LAAO showed no amino acid oxidase activity. Of those, the amino acid oxidase activity of *E. coli* BL21 (pET20b-*Ro*LAAO) and *E. coli* BL21 (pET20b-*Pmi*LAAO) were $1.1 \, U \, ml^{-1}$ and $1.8 \, U \, ml^{-1}$, respectively. Therefore, the *E. coli* BL21 (pET20b-*Pmi*LAAO) showing the highest activity was selected for further study instead of the pET20b-*Ro*LAAO (figure 1).

## 3.2. Preparation of the recombinant *Pmi*LAAO

Different expression vector (pETDuet-1, pCDFduet-1, pET20b, pACYCduet-1 and pRSFduet-1) were selected for *Pmi*LAAO genes expression and were subjected to LAAO activity test. Figure 2*a* shows

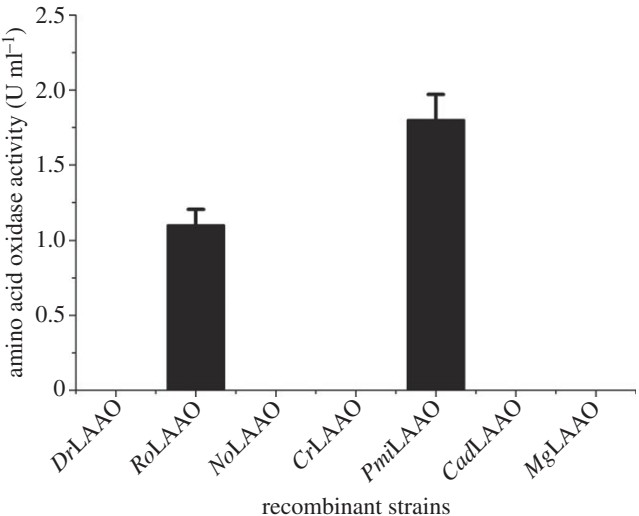

**Figure 1.** The amino acid oxidase (LAAO) activity of the recombinant strains. Amino acid oxidase NoLAAO, CrLAAO, CadLAAO, MgLAAO, RoLAAO, DrLAAO and PmiLAAO were from *Naja oxiana*, *Calloselasma rhodostoma*, *Crotalus adamanteus*, *Meleagris gallopavo*, *Rhodococcus opacus*, *Daboia russellii* and *Proteus mirabilis*, respectively.

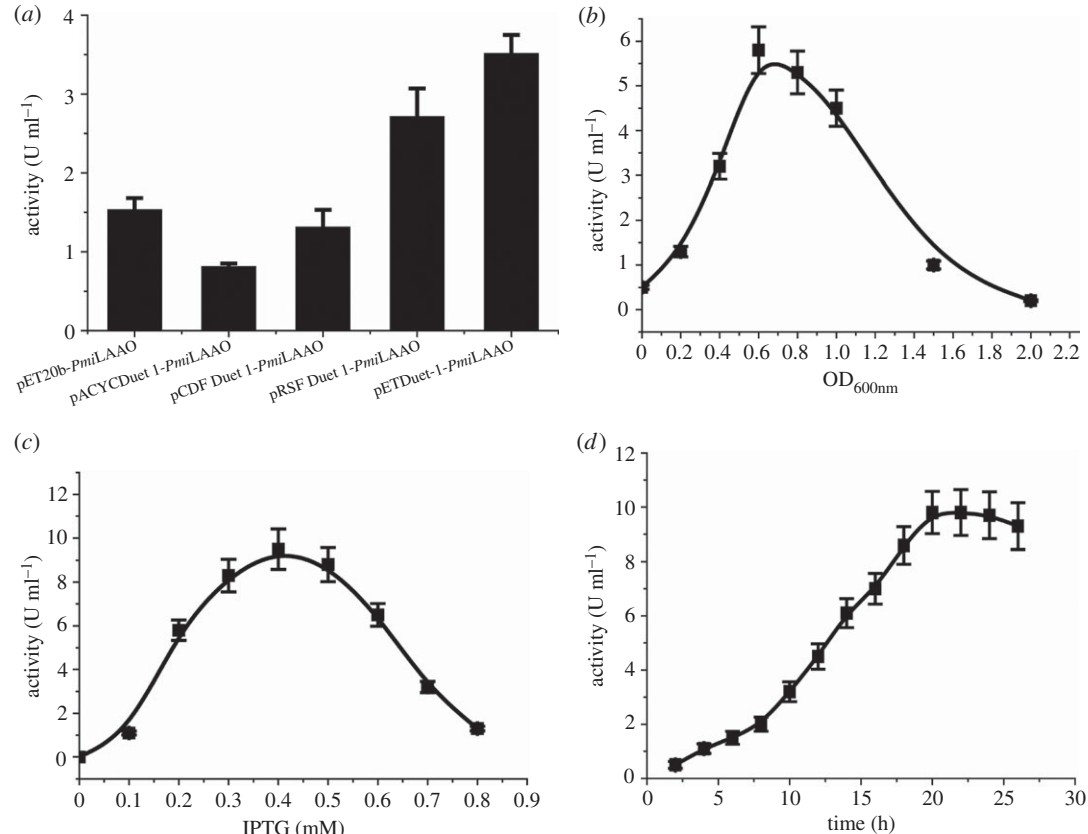

**Figure 2.** Optimization of L-amino acid oxidase PmiLAAO expression. (*a*) Expression vector type, (*b*) cell density for IPTG induction, (*c*) IPTG concentration and (*d*) IPTG induction time on the amino acid oxidase.

that the LAAO activity reached the maximum value ($3.5 \, \text{U ml}^{-1}$) when the PmiLAAO was carried by pETDuet-1 vector. The LAAO activity of pETDuet-1-PmiLAAO was increased by 130% comparing to the pET20b-PmiLAAO. The optimal induction time was at mid-log phase ($OD_{600nm} \sim 0.65$) (figure 2*b*). The optimal IPTG concentration was $0.45 \, \text{mmol l}^{-1}$. Moreover, the maximum PmiLAAO amino acid oxidase activity ($9.8 \, \text{U ml}^{-1}$) was observed after 20 h of induction (figure 2*c,d*).

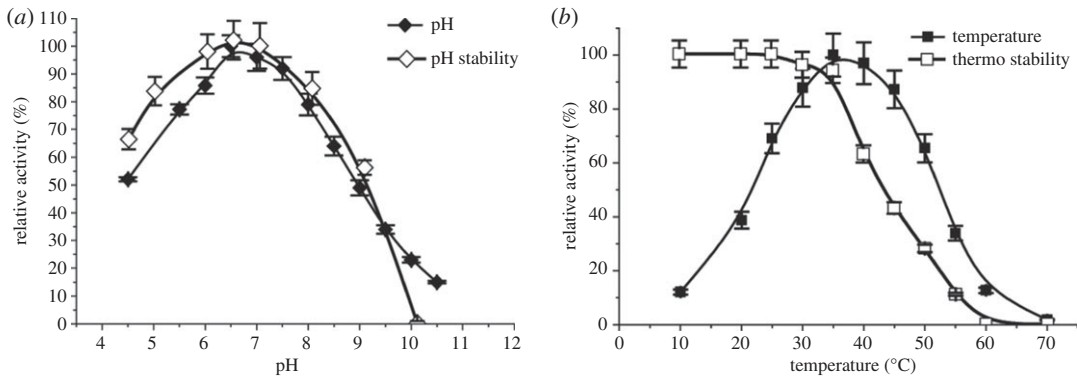

**Figure 3.** Optimal pH and pH stability of L-AAD (*a*). Optimal temperature and thermo stability of L-AAD (*b*). (*a*) (filled diamonds) pH, (open diamonds) pH stability. (*b*) (filled squares) temperature, (open squares) thermostability.

**Table 2.** Purification of *Pmi*LAAO. Note: data are presented as the mean ± s.d. with triple independent measurements.

| steps | total protein (mg) | total activity (mU) | specific activity (mU/mg) | yield (%) | purification fold |
|---|---|---|---|---|---|
| cell lysate (5 l) | 1880 ± 45 | 8900 ± 323 | 4.73 ± 0.166 | 100 | 1.00 |
| ultrafiltration | 432 ± 16 | 6846 ± 240 | 15.83 ± 0.55 | 76.92 | 3.33 |
| HisTrap | 21.5 ± 0.63 | 1822 ± 110 | 84.75 ± 3.43 | 20.47 | 17.92 |

The recombinant *Pmi*LAAO was concentrated by ultrafiltration and purified by affinity chromatography (table 2). The specific activity of *Pmi*LAAO was up to $84.75 \pm 3.43 \, \text{mU mg}^{-1}$ with a purification fold of 17.92 by HisTrap affinity chromatography. In addition, the results show that the optimal pH and temperature of recombinant *Pmi*LAAO were 6.5 and 37°C, respectively. The *Pmi*LAAO was stable in the pH range of 4.0–9.0 and in the temperature range of 10–40°C (figure 3*a*,*b*).

## 3.3. Enhancing the catalytic efficiency of *Pmi*LAAO by immobilization

In order to enhance the α-keto acids production and simplify product separation process, different carriers were used to immobilize the recombinant *E. coli*-pET duet-1-*Pmi*LAAO. The result showed that the method of glutaraldehyde crosslinking and carrageenan entrapment, gelatin entrapment or polyacrylamide gel entrapment was not suitable for the recombinant *E. coli* whole-cell immobilization. Less than 50% of *Pmi*LAAO activity remained after cross-linking with glutaraldehyde, entrapment in carrageenan or entrapment in gelatin and immobilization in polyacrylamide. On the contrary, *E. coli*-pETduet-1-*Pmi*LAAO immobilized in Ca-alginate beads remained 87% activity (figure 4*a*). The immobilized cells further demonstrated improved efficacy by retaining 60% activity even in the seventh reuse cycle (figure 4*b*).

## 3.4. Whole-cell transformation of α-keto acids in packed-bed reactor

The ability of the immobilized whole-cell *E. coli*-pETduet-1-*Pmi*LAAO on resolution of racemate mixtures (D/L-Phe) has been investigated. The greater than 97.5% enantiomeric excess (ee) of (R)-2-amino-2-phenylpropanoic acid (D-Phe) and 49.5% yield of 2-oxo-3-phenylpropanoic acid were observed (figure 5*a*,*b*), respectively. Moreover, for the immobilized whole-cell *E. coli*-pETduet-1-*Pmi*LAAO, the optimum substrate concentration was $10.0 \, \text{g l}^{-1}$. The conversion was decreased, while the substrate concentration was higher than $10.0 \, \text{g l}^{-1}$ (figure 5*c*). The conversion of L-Phe was up to 98%, while the whole-cell biocatalyst loading was $40 \, \text{g l}^{-1}$ (figure 5*d*). In addition, the changes of conversion were not significant by further increasing the whole-cell biocatalyst loading. After conditional optimization, $10.0 \, \text{g l}^{-1}$ L-phenylalanine was almost completely transformed to 2-oxo-3-phenylpropanoic acid with a conversion of 99.5% (figure 5*e*).

It was shown that the conversion did not increase along with the increasing substrate concentration (figure 5*c*), which indicated the immobilized *E. coli*-pETduet-1-*Pmi*LAAO caused substrate inhibition. In

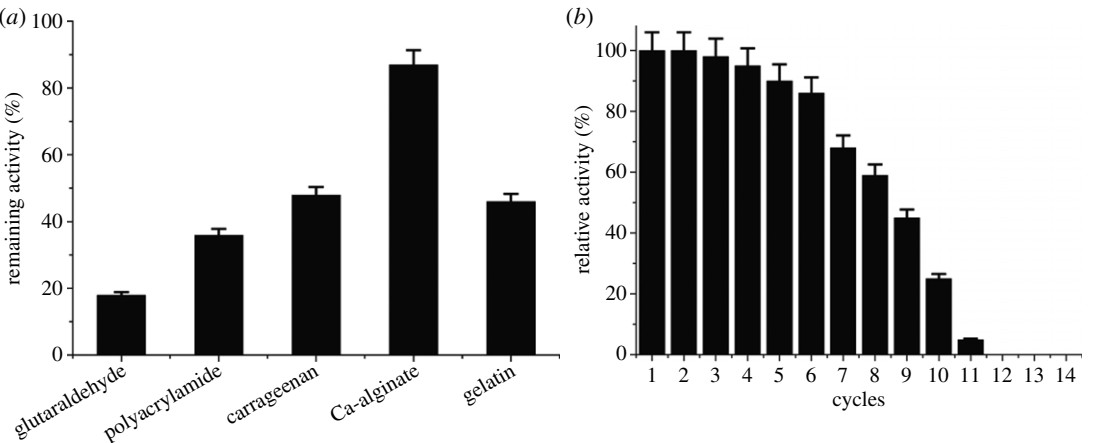

**Figure 4.** Immobilization of the whole-cell recombinant *E. coli* (*a*). Reusability of immobilized whole cells (*b*). Remaining activity = (*Pmi*LAAO activity before immobilization-*Pmi*LAAO activity after immobilization)/*Pmi*LAAO activity before immobilization × 100%.

**Figure 5.** Preparation of chiral amino acids and α-keto acids by immobilized *E. coli*-pETduet-1-*Pmi*LAAO with a 4 mM substrate (*a*). Racemic resolution of 2-amino-2-phenylpropanoic acid by *Pmi*LAAO (*b*). Solutions of 16 mM (2.64 g l$^{-1}$) (ᴅ/ʟ)-phenylalanine with 5 g l$^{-1}$ (wet cell weight, WCW) immobilized *E. coli*-pETduet-1-*Pmi*LAAO in 200 mM phosphate buffer (pH = 6.5) were incubated at 37°C with gentle shaking. Samples were withdrawn regularly, and the product were analysed by HPLC. Effect of ʟ-phenylalanine concentration on α-keto acids production by immobilized *E. coli*-pETduet-1-*Pmi*LAAO (*c*); effect of the whole-cell biocatalyst loading on α-keto acids production (*d*); time curve of 2-oxo-3-phenylpropanoic acid production from ʟ-phenylalanine (10 g l$^{-1}$) by immobilized *E. coli*-pETduet-1-*Pmi*LAAO(E).

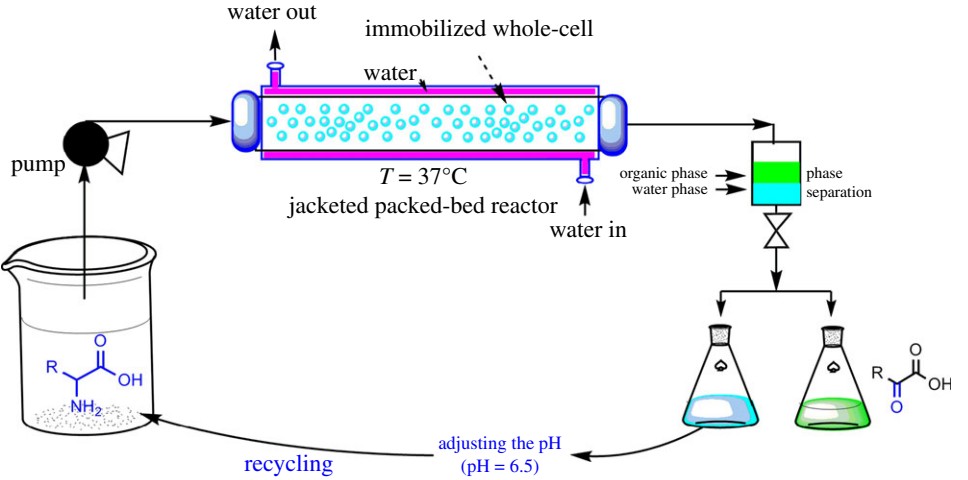

**Figure 6.** Continuous biosynthesis of *2-oxo-3-phenylpropanoic acid* from L-Phe by immobilized whole-cell of *E. coli*-pETduet-1-*Pmi*LAAO in packed-bed reactor. The pH of the aqueous phase containing unreacted amino acids (substrate) had been adjusted to 6.5 before recycling.

**Table 3.** Continuous production of 2-oxo-3-phenylpropanoic acid from L-phenylalanine. Note: the aqueous phase containing an amount of unreacted substrate (L-phenylalanine) obtained from product was used to react again.

| L-phenylalanine con. (g $l^{-1}$) | flow rate (ml $h^{-1}$) | residence time (h) | 2-oxo-3-phenylpropanoic acid con. (g $l^{-1}$) | conversion (%) | space–time yield (g $l$ $h^{-1}$) |
|---|---|---|---|---|---|
| 5 | 10 | 25.00 | 4.96 | 99.9 | 0.20 |
| | 20 | 12.50 | 4.92 | 99 | 0.39 |
| | 40 | 6.25 | 4.40 | 88.5 | 0.70 |
| | 60 | 4.17 | 3.62 | 72.8 | 0.87 |
| | 80 | 3.13 | 3.23 | 65 | 1.03 |
| | 100 | 2.50 | 2.65 | 53.4 | 1.06 |
| 10 | 20 | 12.50 | 9.16 | 92.2 | 0.73 |
| 20 | 20 | 12.5 | 15.40 | 77.5 | 1.23 |
| 30 | 20 | 12.5 | 18.72 | 62.8 | 1.50 |
| 40 | 20 | 12.5 | 17.69 | 44.5 | 1.42 |
| 50 | 20 | 12.5 | 13.02 | 26.2 | 1.04 |
| 60 | 20 | 12.5 | 7.87 | 13.2 | 0.63 |
| 30 | 20 | 25 | 29.66 | 99.5 | 1.19 |

order to enhance the 2-oxo-3-phenylpropanoic acid production, immobilized *E. coli*-pETduet-1-*Pmi*LAAO was filled into the jacketed tubular reactor (figure 6). Along with the increasing flow rate, the substrate conversion decreased and the space–time yield increased while the L-phenylalanine concentration was 5 g $l^{-1}$ (table 3). Although the conversion was close to 100%, the lowest space–time yield of 0.2 g $l$ $h^{-1}$ was obtained at a residence time of 25 h (5 g $l^{-1}$ L-phenylalanine). In order to maintain a high substrate conversion rate, the residence time (12.50 h) and flow rate (20 ml $h^{-1}$) was selected for further study.

Although the conversion was decreased along with the increasing substrate concentration while the L-phenylalanine concentration was 5 g $l^{-1}$, higher productivity could be enhanced by increasing the substrate concentration. Significantly, the maximal space–time yield of 1.50 g $l$ $h^{-1}$ was achieved at a residence time of 12.5 h (30 g $l^{-1}$ L-phenylalanine). A concentration of 29.66 g $l^{-1}$ 2-oxo-3-phenylpropanoic acid with a substrate conversion rate of 99.5% was achieved by correspondingly increasing the residence time (25 h) under fixed substrate concentration of 30 g $l^{-1}$.

**Table 4.** Comparison of 2-oxo-3-phenylpropanoic acid production efficiency. Note: dash indicates that it is not stated in the text.

| | 2-oxo-3-phenylpropanoic acid con. (g l$^{-1}$) | conversion (%) | space–time yield (g l h$^{-1}$) | ref. |
|---|---|---|---|---|
| chemical synthesis | — | 50 (yield) | — | [22] |
| fermentation | 1.054 | 80 | 0.0337 | [23] |
| D-amino acid oxidase from porcine kidney | 0.23 | 55 | 0.038 | [24] |
| coimmobilized D-amino acid oxidase/ catalase | 3.304 | 50 | 1.802 | [25] |
| PmiLAAO | 0.75 | 75 | 0.1 | this study |
| immobilized whole-cell | 1.31 | 99.8 | 0.66 | this study |
| immobilized whole-cell + packed-bed reactor | 29.66 | 99.5 | 1.19 | this study |
| pure enzyme | 2.6 | 86.7 | 1.04 | [21] |
| whole-cell | 3.3 | 82.5 | 0.55 | [21] |
| LAAO-D165 K/F263 M/L336 M + substrate feeding | 22.8 | 68 | 2.85 | [4] |

# 4. Discussion

In conclusion, an L-amino acid oxidase (PmiLAAO) from *Proteus mirabilis* has been cloned and expressed in *E. coli*. pETDuet-1-PmiLAAO activity improved by expression vector screening and expression condition optimization ($OD_{600} = 0.65$, 0.45 mmol l$^{-1}$ IPTG, 20 h of induction). Afterwards, *E. coli*-pETduet-1-PmiLAAO was immobilized in Ca-alginate beads to simplify product separation process. Therefore, continuous production of 2-oxo-3-phenylpropanoic acid was conducted in a packed-bed reactor via immobilized *E. coli*-pETduet-1-PmiLAAO. Significantly, 99.5% substrate conversion rate was achieved by correspondingly increasing the residence time.

Biocatalysis has been regarded as an efficient and economical method for preparation of α-keto acids and optically chiral amino acids under mild conditions. However, most biotransformations are short of ideal catalyst to meet the requirement of industrial application [17–19]. Enzymes with high selectivity and specific activity are an important prerequisite for industrial application. In the previous study, α-keto acids, key intermediates for the preparation of D-amino acids and optically active α-hydroxyl carboxylic acids had been prepared by amino acid oxidases, however, only a few biocatalysts were available. Therefore, in order to obtain an usable and efficient L-amino acid oxidase, enzyme database searching was performed based on BLAST, which has been proved to be a time-saving approach, compared with the strain screening from soil samples. Several LAAO with catalytic activity towards L-Phe have been identified in this study. Unfortunately, the DrLAAO from *Daboia russellii* show no activity in present study, probably due to DrLAAO being from eukaryotes, and DrLAAO fail to be properly assembled and folded in *E. coli*. By gene expression vector screening and expression condition optimization, the expression level of PmiLAAO has been enhanced in this study. Furthermore, in order to enhance the α-keto acids production and simplify product separation process, *E. coli*-pETduet1-PmiLAAO was immobilized in Ca-alginate beads. Notably, the full resolution (ee > 99%) was also achieved from racemate. Cell immobilization methods are shown as follow: (i) cross-linking of enzyme molecules, (ii) entrapment in polymer matrices and (iii) binding to a prefabricated support [20]. However, the cross-linking with glutaraldehyde, entrapment in carrageenan or gelatin and immobilized in polyacrylamide failed to enhance the activity of PmiLAAO with a 50% activity loss in this study; it is illustrated that the Ca-alginate beads provide a gentle environment for immobilization for PmiLAAO. The whole-cell of the *E. coli*-pETduet1-PmiLAAO was used to prepare the α-keto acids, the conversion of the 10 g l$^{-1}$ L-phenylalanine was up to 99.5% with a working time of 80 min. The catalytic performance of *E. coli*-pETduet1-PmiLAAO whole cell is

higher than L-amino acid oxidase (L-AAO) from *Rhodococcus opacus*, which has been used for conversion of L-4-chlorophenylalanine with a low activity [21]. Afterwards, *E. coli*-pETduet1-*Pmi*LAAO holds the potential to be used for industrial preparation of α-keto acids. Comparing the report of *Ying Hou* [4], the production of 2-oxo-3-phenylpropanoic acid was increased by 23.12% with a conversion of 99.5%, however, the space–time yield was reduced 58.24% (table 4).

Although the production of α-keto acids could be increased by increasing the substrate concentration, the substrate conversion was inefficient [26]. The substrate conversion hindering the α-keto acids preparation as a key factor has been observed in this study. Substrate conversion reduced along with the increasing substrate concentration was observed in this study. In order to further increase the yield of α-keto acids, we are trying our best to eliminate the product inhibition and enhance the substrate affinity by means of protein engineering.

Data accessibility. All data are included in the article. We have conducted our experiments systematically and reported their experimental procedure clearly in the experimental section and provided all the necessary data in the results and discussion section in the main manuscript.

Authors' contributions. Z.L., P.L. and L.W. developed the ideas. L.W., G.W. and P.L. carried out the measurements and participated in data analysis. L.W. and X.G. wrote the manuscript. All authors commented, reviewed and gave final approval for publication.

Competing interests. The authors declare we have no competing interests.

Funding. There is no government or academic funding to support this research.

Acknowledgements. We thank zhejiang zhengshuo Biological Co., Ltd for help.

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
