## [Reviewer comments · Royal Society Open Science]

Review History

RSOS-182035.R0 (Original submission)

Review form: Reviewer 1

Is the manuscript scientifically sound in its present form?

Yes

Are the interpretations and conclusions justified by the results?

Yes

Is the language acceptable?

Yes

Is it clear how to access all supporting data?

Not Applicable

Do you have any ethical concerns with this paper?

No

Have you any concerns about statistical analyses in this paper?

No

Recommendation?

Major revision is needed (please make suggestions in comments)

Comments to the Author(s)

The manuscript by Liu et al. majorly studied the production of α -keto acids using whole cells entrapped in calcium alginate. Firstly, the well expressed recombinant strains were selected by the comparison of different plasmids. Then the authors focused on the characterization of PmiLAAO and preparation of α -keto acids from L-amino acid by the whole-cell biocatalyst. Although 29.66 g/L of 2-oxo-3-phenylpropanoic acid was finally produced from the reactor, the yield was still relatively low. Therefore, how to further increase the yield of α -keto acids? Overall this manuscript represents a comprehensive study in the field of whole-cell biocatalysis. It is acceptable for publication with some careful corrections. Some detail suggestions were given as follows.

1. Fig. 3 showed that optimal pH of the recombinant enzyme was 6.5. However, as shown in Scheme 1, NH₃ produced from the reaction may change the pH of this particular system. Discussion and explanation are needed.
2. In Line 25 and 28, errors of article style appeared and need to be corrected.
3. In Line 148, 2 g/L of whole cell biocatalyst might be too little in quantity. It would be persuasive if the experimental data of optimal wet cell weight was given.
4. In Table 1, the Km of PmiLAAO was the highest in the list. Accordingly, low Km means high affinity for substrate. The authors need to give more discussion to explain the reason of choosing PmiLAAO.
5. In Line 188 and 271, the same error appeared as comment 2 and needs to be corrected.

Review form: Reviewer 2

Is the manuscript scientifically sound in its present form?

Yes

Are the interpretations and conclusions justified by the results?

Yes

Is the language acceptable?

Yes

Is it clear how to access all supporting data?

Yes

Do you have any ethical concerns with this paper?

No

Have you any concerns about statistical analyses in this paper?

No

Recommendation?

Accept with minor revision (please list in comments)

Comments to the Author(s)

The authors demonstrated simple continuous bio-approach for effective production of α -keto acids by immobilized *E. coli*-pETduet-1-PmiLAAO in a jacketed packed-bed reactor. The recycling experiments and different substrates study added the value to the work. The manuscript is well written, systematic, and quality of the article is good in my opinion. After reading the whole manuscript, I suggest acceptance of the manuscript after the following minor revisions:

- 1) In introduction, in line 43, authors should provide abbreviation of TCA cycle, necessary for common readers.
- 2) Table 4. Comparison of 2-oxo-3-phenylpropanoic acid production efficiency, authors can supply more examples from the literature.

Decision letter (RSOS-182035.R0)

25-Jan-2019

Dear Dr Liu:

Title: Efficient Production of α -keto acids by immobilized *E. coli*-pETduet-1-PmiLAAO in a jacketed packed-bed reactor
Manuscript ID: RSOS-182035

The editor assigned to your manuscript has now received comments from reviewers. We would like you to revise your paper in accordance with the referee and Subject Editor suggestions which can be found below (not including confidential reports to the Editor). Please note this decision does not guarantee eventual acceptance.

Please submit your revised paper before 17-Feb-2019. Please note that the revision deadline will expire at 00.00am on this date. If we do not hear from you within this time then it will be assumed that the paper has been withdrawn. In exceptional circumstances, extensions may be possible if agreed with the Editorial Office in advance. We do not allow multiple rounds of revision so we urge you to make every effort to fully address all of the comments at this stage. If deemed necessary by the Editors, your manuscript will be sent back to one or more of the original reviewers for assessment. If the original reviewers are not available we may invite new reviewers.

Please also include the following statements alongside the other end statements. As we cannot publish your manuscript without these end statements included, if you feel that a given heading is not relevant to your paper, please nevertheless include the heading and explicitly state that it is not relevant to your work.

• Ethics statement

Please clarify whether you received ethical approval from a local ethics committee to carry out your study. If so please include details of this, including the name of the committee that gave consent in a Research Ethics section after your main text. Please also clarify whether you received informed consent for the participants to participate in the study and state this in your Research Ethics section.

OR

Please clarify whether you obtained the necessary licences and approvals from your institutional animal ethics committee before conducting your research. Please provide details of these licences and approvals in an Animal Ethics section after your main text.

OR

Please clarify whether you obtained the appropriate permissions and licences to conduct the fieldwork detailed in your study. Please provide details of these in your methods section.

RSC Associate Editor:
Comments to the Author:
(There are no comments.)

RSC Subject Editor:
Comments to the Author:
(There are no comments.)

Reviewers' Comments to Author:

Reviewer: 1

Comments to the Author(s)

The manuscript by Liu et al. majorly studied the production of α -keto acids using whole cells entrapped in calcium alginate. Firstly, the well expressed recombinant strains were selected by the comparison of different plasmids. Then the authors focused on the characterization of PmiLAAO and preparation of α -keto acids from L-amino acid by the whole-cell biocatalyst. Although 29.66 g/L of 2-oxo-3-phenylpropanoic acid was finally produced from the reactor, the yield was still relatively low. Therefore, how to further increase the yield of α -keto acids? Overall this manuscript represents a comprehensive study in the field of whole-cell biocatalysis. It is acceptable for publication with some careful corrections. Some detail suggestions were given as follows.

1. Fig. 3 showed that optimal pH of the recombinant enzyme was 6.5. However, as shown in Scheme 1, NH₃ produced from the reaction may change the pH of this particular system. Discussion and explanation are needed.
2. In Line 25 and 28, errors of article style appeared and need to be corrected.
3. In Line 148, 2 g/L of whole cell biocatalyst might be too little in quantity. It would be persuasive if the experimental data of optimal wet cell weight was given.
4. In Table 1, the Km of PmiLAAO was the highest in the list. Accordingly, low Km means high affinity for substrate. The authors need to give more discussion to explain the reason of choosing PmiLAAO.
5. In Line 188 and 271, the same error appeared as comment 2 and needs to be corrected.

Reviewer: 2

Comments to the Author(s)

The authors demonstrated simple continuous bio-approach for effective production of α -keto acids by immobilized E. coli-pETduet-1-PmiLAAO in a jacketed packed-bed reactor. The recycling experiments and different substrates study added the value to the work. The manuscript is well written, systematic, and quality of the article is good in my opinion. After reading the whole manuscript, I suggest acceptance of the manuscript after the following minor revisions:

- 1) In introduction, in line 43, authors should provide abbreviation of TCA cycle, necessary for common readers.
- 2) Table 4. Comparison of 2-oxo-3-phenylpropanoic acid production efficiency, authors can supply more examples from the literature.

Author's Response to Decision Letter for (RSOS-182035.R0)

See Appendix A.

RSOS-182035.R1 (Revision)

Review form: Reviewer 1

Is the manuscript scientifically sound in its present form?

Yes

Are the interpretations and conclusions justified by the results?

Yes

Is the language acceptable?

Yes

Is it clear how to access all supporting data?

Yes

Do you have any ethical concerns with this paper?

No

Have you any concerns about statistical analyses in this paper?

No

Recommendation?

Accept as is

Comments to the Author(s)

The authors have addressed my concerns.

Review form: Reviewer 2

Is the manuscript scientifically sound in its present form?

Yes

Are the interpretations and conclusions justified by the results?

Yes

Is the language acceptable?

Yes

Is it clear how to access all supporting data?

Yes

Do you have any ethical concerns with this paper?

No

Have you any concerns about statistical analyses in this paper?

No

Recommendation?

Accept as is

Comments to the Author(s)

Comments:

Manuscript seems okay now. It can be accepted as it is.

Decision letter (RSOS-182035.R1)

01-Mar-2019

Dear Dr Liu:

Title: Efficient Production of α -keto acids by immobilized *E. coli*-pETduet-1-PmiLAAO in a jacketed packed-bed reactor
Manuscript ID: RSOS-182035.R1

It is a pleasure to accept your manuscript in its current form for publication in Royal Society Open Science. The chemistry content of Royal Society Open Science is published in collaboration with the Royal Society of Chemistry.

RSC Associate Editor:
Comments to the Author:
(There are no comments.)

RSC Subject Editor:
Comments to the Author:
(There are no comments.)

Reviewer(s)' Comments to Author:

Reviewer: 2

Comments to the Author(s)

Comments:

Manuscript seems okay now. It can be accepted as it is.

Reviewer: 1

Comments to the Author(s)

The authors have addressed my concerns.

Appendix A

Dear Chief editor

Thank you very much for your kind letter and encouragement, along with the positive and constructive comments of reviewers concerning our manuscript (Manuscript ID: RSOS-182035).

We have substantially revised our manuscript, after thoroughly considered all the comments and suggestions of reviewers and editor. Ethics statement has been added in the revised manuscript (page 12, Line 300-301). The point-to-point answers and explanations for all revisions were listed in a separate paper following this letter. All the revisions are highlighted in yellow throughout the revised manuscript.

We hope, with these modifications and improvements based on your suggestions and the reviewer's comments, the quality of our manuscript would meet the publication standard of **Royal Society Open Science**.

If you have any question about this paper, please contact us; Thank you for considering this work and look forward to your response.

With kind personal regards,

Sincerely yours,

Ziduo Liu

College of Life Science and Technology, State Key Laboratory of Agricultural Microbiology, Huazhong Agricultural University, Wuhan 430070, China. Tel.: +86 27 87281429. E-mail: ziduoliu_hzauedu@sina.cn

For your guidance, itemized answers to reviewer's comments is appended below (answers are in blue font).

Reviewer: 1

(1) The manuscript by Liu et al. majorly studied the production of α -keto acids using whole cells entrapped in calcium alginate. Firstly, the well expressed recombinant strains were selected by the comparison of different plasmids. Then the authors focused on the characterization of PmiLAAO and preparation of α -keto acids from L-amino acid by the whole-cell biocatalyst. Although 29.66 g/L of 2-oxo-3-phenylpropanoic acid was finally produced from the reactor, the yield was still relatively low. **Therefore, how to further increase the yield of α -keto acids?** Overall this manuscript represents a comprehensive study in the field of whole-cell biocatalysis. It is acceptable for publication with some careful corrections.

A: Thanks for your kind suggestion. We speculate that there may be a substrate inhibition or product inhibition hindering the improvement of α -keto acids yield. In order to further increase the yield of α -keto acids, we are trying our best to eliminate the product inhibition and enhance the substrate affinity by means of protein engineering. The work of *pmiLAAO* gene mutation based on molecular docking and molecular dynamics is underway. "In order to further increase the yield of α -keto acids, we are trying our best to eliminate the product inhibition and enhance the substrate affinity by means of protein engineering." has been added in the revised manuscript (Page 12, Line 285-287).

(2) Fig. 3 showed that optimal pH of the recombinant enzyme was 6.5. However, as

shown in Scheme 1, NH₃ produced from the reaction may change the pH of this particular system. Discussion and explanation are needed.

A: Thanks for your kind suggestion. The pH of the aqueous phase containing unreacted amino acids (substrate) had been adjusted to 6.5 before recycling. The step of pH adjusting has been added into the Revised Figures and Tables (Page 1, Scheme 1; Page 4, Fig.6). "The pH of the aqueous phase containing unreacted amino acids (substrate) had been adjusted to 6.5 before recycling." has been added in the Revised Figures and Tables (Page 1, Scheme 1; Page 4, Fig.6).

(3) In Line 25 and 28, errors of article style appeared and need to be corrected.

A: Thanks for your kind suggestion. Errors of article style has been corrected in the revised manuscript (Page 1-2, Line 25-28), including "130 %" has been changed to "130%"; "5–40 °C " has been changed to "5–40°C "; " pH of 5.0 - 8.0" has been changed to " pH of 5.0-8.0"; "Furthermore, the PmiLAAO was stable at 5–40°C and pH of 5.0-8.0, the optimal pH and temperature of recombinant PmiLAAO were 6.5 and 37°C, respectively." has been changed to "Furthermore, The *PmiLAAO* was stable in the pH range of 4.0–9.0 and in the temperature range of 10–40°C; The optimal pH and temperature of recombinant PmiLAAO were 6.5 and 37°C, respectively"

(4) In Line 148, 2 g/L of whole cell biocatalyst might be too little in quantity. It would be persuasive if the experimental data of optimal wet cell weight was given.

A: Thanks for your kind suggestion. 2 g/L of whole cell biocatalyst was used for functional screening of the biocatalysts. The experimental data of optimal whole-cell biocatalyst loading has been added in the Revised Figures and Tables

(Page 3, Fig. 5 D). "The conversion of L-Phe was up to 98 %, while the whole-cell biocatalyst loading was 40 g/L (Fig. 5D)" has been added in the revised manuscript (Page 9, Line 212-215).

- (5) In Table 1, the K_m of *PmiLAAO* was the highest in the list. Accordingly, low K_m means high affinity for substrate. The authors need to give more discussion to explain the reason of choosing *PmiLAAO*.

A: Thanks for your kind suggestion. K_m value was obtained from Enzyme Database -BRENDA (measured by different researchers). In the present study, the K_m values was just used as reference to obtain the source of LAAO." However, many of LAAO gene sequences with lower K_m value (including *NcLAAO*, *PreLAAO*, *ThLAAO*, *HsLAAO*, *CatLAAO*) were failed to be accessed from the GenBank database (<https://www.ncbi.nlm.nih.gov/genbank/>). *NoLAAO*, *CrLAAO*, *CadLAAO*, *MgLAAO*, *PmiLAAO*, *RoLAAO* and *DrLAAO* with available sequence data were selected as the candidates. Afterwards, pET20b-*NoLAAO*, pET20b-*CrLAAO*, pET20b-*CadLAAO*, pET20b-*MgLAAO*, pET20b-*PmiLAAO*, pET20b-*DrLAAO*, and pET20b-*RoLAAO* were constructed and transformed into the expression host *E. coli* BL21 (DE3), subsequently." has been added in the revised manuscript (Page 7, Line 161-169). "Unfortunately, the *NoLAAO*, *CrLAAO*, *CadLAAO*, *MgLAAO*, and *DrLAAO* showed no amino acid oxidase activity. Of those, the amino acid oxidase activity of *E. coli* BL21 (pET20b -*RoLAAO*) and *E. coli* BL21 (pET20b -*PmiLAAO*) were 1.1 U/mL and 1.8 U/mL, respectively. Therefore, the *E. coli* BL21 (pET20b -*PmiLAAO*) showing highest activity was selected for further study instead of the pET20b -*RoLAAO* (Fig. 1)." has been added in the revised manuscript (Page 7, Line 170-175).

(6) In Line 188 and 271, the same error appeared as comment 2 and needs to be corrected.

A: Thanks for your kind suggestion. Errors of article style has been corrected in the revised manuscript (Page 8-11, Line 188-271), including "37 °C " has been changed to "37°C "; " 50 %" has been changed to " 50%"; "> 97.5 % enantiomeric excess (ee)" has been changed to " > 97.5% enantiomeric excess (ee)"; " (ee > 99 %)" has been changed to "(ee>99%)"; " The *Pmi*LAAO was stabile within a pH range of 4.0–9.0 and at a temperature range of 10–40°C " has been changed to " The *Pmi*LAAO was stabile in the pH range of 4.0–9.0 and in the temperature range of 10–40°C"(Page 8, Line 188-189).

Reviewer: 2

(1) In introduction, in line 43, authors should provide abbreviation of TCA cycle, necessary for common readers.

A: Thanks for your kind suggestion. The abbreviation of TCA cycle has been provided in the revised manuscript (Page 2, Line 43-45). "... group in a molecule involved in the TCA cycle (tricarboxylic acid cycle, also well known as citric acid cycle), decomposition of amino acids and significant in biological systems "

(2) Table 4. Comparison of 2-oxo-3-phenylpropanoic acid production efficiency, authors can supply more examples from the literature.

A: Thanks for your kind suggestion. more examples from the literature has been supplied to the revised Figures and Tables (Page 5, Table 4), including Chemical synthesis, Fermentation, D-amino acid Oxidase from porcine kidney, Coimmobilized D-amino acid oxidase /catalase